# Assessing Uranium Pollution Levels in the Rietspruit River, Far West Rand Goldfield, South Africa

**DOI:** 10.3390/ijerph18168466

**Published:** 2021-08-11

**Authors:** Iyioluwa Busuyi Raji, Emile Hoffmann, Adeline Ngie, Frank Winde

**Affiliations:** Unit of Environmental Science and Management, Faculty of Natural and Agricultural Sciences, Potchefstroom Campus, North-West University, Private Bag X6001, Potchefstroom 2520, South Africa; emile.hoffmann@nwu.ac.za (E.H.); Adeline.Ngie@nwu.ac.za (A.N.); frank.winde@gmail.com (F.W.)

**Keywords:** uranium, pollution, wetland, dam, water salinity

## Abstract

The Rietspruit is located in Gauteng Province, South Africa, within the Witwatersrand Basin. The basin is noted for its vast gold deposit. The river extends for about 60 km before joining the Vaal River. The aim of this study was to determine the concentration of uranium in the Rietspruit and the factors that influenced the concentration of U at each of the sites. The source of uranium in the river is the discharge from the gold mine and the mine wastes. Inductively coupled plasma mass spectrometry was used for water and sediment analysis in order to determine the concentration of U. High concentration of U was found in the river water and sediment, which is above the permissible limit of U. The water is used for irrigation of farmlands, cattle watering and for human consumption despite the high concentration of uranium in it. Ingestion of uranium is dangerous to human health. Due to the toxic nature of uranium, consumption of the water for domestic use and agriculture purpose must be discouraged.

## 1. Introduction

Uranium (U) is an isotopic radioactive element [1]. This is because of the difference in the number of neutrons in the nucleus of the atom. The natural mean background concentration of U in soil is ≤4 mg/kg [2], and the South African limit of U in water used for drinking and agricultural activities is 30 µg/L. However, gold mining operations elevated the concentration of U in sediment and water [3]. In South Africa, U is found in association with gold, and South African uranium is mainly produced as a by-product of gold mining [4].

The mining of gold brought a large concentration of U to the surface. In South Africa, about six million kilograms of U were brought to the surface and disposed off onto slime dams [5]. Slime dams contaminate nearby surface water as well as ground water through different modes of erosion (water and air) and seepage to water systems [6,7]. The contamination of water bodies is hazardous to humans and animals that depend on the water for various uses [8,9,10,11]. Due to the mobility of U influenced by the redox state of U, chemistry of the water and sediment as well as iron and manganese concentration in the sediment [12,13], downstream users of the water could also be exposed. Exposure pathways of human to U include drinking of polluted water, consumption of food rich in U, inhalation of mine dusts and farm produce irrigated with contaminated water [14,15,16].

Several studies have been done on the impacts of gold mining in South Africa focusing on U [6,15,17,18] reporting high concentration of U within gold mines; however, no literature has reported on the concentration of U for the entire length of the Rietspruit. As a result of this, it is important to know the concentration of U in the Rietspruit system (water and sediment), and also the extent of the pollution plus the factors influencing U concentration. The use of water for irrigation provides a pathway for the introduction of U into the food chain, and also, drinking the water as a religious rite also makes it crucial to quantify the level of the pollutant in the water. This is necessary in order to engage the authority in charge of water to take adequate steps in reducing human exposure. This study focused on the U pollution level of the Rietspruit from the Peter Wright (PW) dam downstream to the Vaal River about 60 km and the factors that played role(s) on the U level.

The main objective of this study was to determine the concentration of U in the Rietspruit water and sediment. Factors that play an important role in the mobilization and immobilization of U will also be discussed.

## 2. Materials and Methods

### 2.1. Study Area

The Rietspruit River is located in the Highveld region of South Africa. It starts from Peter Wright dam, a reservoir located at an elevation of 1680 m above sea level, in the West Rand of Gauteng (C22H Quaternary Catchment area according to the Department of Water Affairs and Forestry) and flows for about 60 km before the confluence with the Vaal River (Figure 1C). The Vaal River is located downstream of Vanderbijlpark and extends up to the Northern Cape, where it also joins the Orange River. As a result of the abundance of gold deposits in this study area, the Rietspruit area has experienced over 60 years of gold mining activity. Notable mines include Cooke 4 Ezulwini Plant and South Deep Gold Mine, which is along the Leeuspruit River (Figure 1C). The Leeuspruit River formed a confluence with the Rietspruit River, and it is found in the C22J Quaternary Catchment area. The total size of these two catchments is about 1123 km^2^. Other notable tributaries include the Evaton Rietspruit and Klein Rietspruit (Figure 1C).

The study area is within the Witwatersrand Basin, an Archean sedimentary basin that contains a stratigraphic sequence about 6 km thick. It is made up of quartzites and shales with minor volcanic unit. Gold is hosted by the upper Elsburg and middle Elsburg reefs. Uranium is also found in the middle Elsburg reef. The gold is found associated with sulphide minerals (pyrite), while U is found in the form of uraninite.

### 2.2. Field Work

In total, thirty soil samples and twenty-eight water samples were collected from the 30 sites downstream of the Rietspruit at varying distances, depending on accessibility while focusing on wetlands, dams, river confluence and where human exposures were witnessed. Sediments were collected at the same spot where water samples were collected using an auger. Sampling started in the vicinity of the gold mine and continued downstream (Figure 1C). Water samples were collected under the water surface using water bottles in the middle of the river. Water bottles were rinsed three times before taking water samples. This is done to avoid sampled water contamination. When the water bottle was completely filled, the bottle was capped under water. There were two dry samples from the tailings; each from the slime dam and mine waste located along the furrow leading to the PW dam inflow, while the rest of the sediment samples were collected under water at the same spot where water samples were collected. The sediment samples were kept in a zip-lock bag, while the water samples were kept in the 500 mL water bottle used to collect the water samples. The pH, electric conductivity and temperature of the water body were taken using the WTW multi 350i at each of the sampled sites (reference temperature: 25 °C). Stream flow velocity was measured using Global Flow Probe FP101 and 201. All samples were taken to the laboratory after sampling.

### 2.3. Lab Analyses

Sediment samples were all dried in the oven at a temperature of 50 °C for 24 h to remove the water content. About 400 g of the sediment was placed in a labelled petri dish. The petri dish was then arranged in the oven. After drying, a mortar and pestle were used to break the particles and a sieve of mesh size 2000 µm was used to remove gravel and plant particles. Sediment samples were digested according to EPA 3050b. About 200 mg of the sieved sediment sample were placed in a Teflon tube. A volume of 9 mL of 65% nitric acid and 3 mL of 32% hydrochloric acid was added to the tube, capped and placed in a microwave digestion system (Milestone, Ethos UP, Maxi 44). A period of 20 min allowed the system to reach 1800 MW at a temperature of 200 °C, which was maintained for another 15 min. After cooling, Agilent 7500 CE ICP-MS fitted with Collision Reaction Cell technology for interface removal was used to determine the concentration of U and other trace metals in the samples. In order to achieve a quantitative result, the instrument is calibrated using ULTRASPEC certified custom mixed multi-element stock standard (De Bruyn Spectroscopic Solutions, Midrand, South Africa) solutions containing all the elements of interest.

Water samples were filtered using a 0.45 µm pre-filter nylon sieve to remove suspended particles. Then, 1 mL of the filtered water was mixed with 9 mL of 2% HNO_3_ to remove organic constituents before ICP-MS was used to determine U concentration. High-performance ion chromatography was used to determine the anion concentration of chlorine (Cl), sulphate (SO_4_), nitrate (NO_3_), nitrite (NO_2_), phosphate (PO_4_) and fluorine (F) in the water.

## 3. Results and Discussion

### 3.1. Rietspruit Water Chemistry

The quality of water of the Rietspruit River from twenty-eight (28) water samples is given in Table 1.

### 3.2. pH Value

The pH of the stream water is weakly acidic to weakly alkaline (6.0–8.8) (Figure 2). This is within the WHO guideline range (6–9) and also the South African limit (5–9.7). The temperature at which pH reading are taken should always be reported, since pH measurement is influenced by temperature. The maximum temperature recorded was 26 °C at the channel water, while the minimum temperature was 8.8 °C, which corresponded to the lowest pH reading observed in the field.

The pH of the stream water increases as the sun rises from the first site, inflow to Peter Wright dam, downstream. The maximum pH reading, 8.8, was recorded around noon, when the sun was at its peak (Figure 2). The fluctuation of pH in stream water is as a result of photosynthesis of aquatic plants and algae. The minimum pH reading, 6.0, was recorded in the evening, after sunset. As a result of the sunset, sampling was continued the next day and the trend of the rising pH reading as the sun rises was observed the following day when the last few sites were visited. This is similar to the finding of Winde et al. [21]. However, in his study, pH observation was done in one location, but the pH observations correspond with this study finding.

The reduced species (U(lV), U^4+^) typically precipitates from natural waters near neutral pH due to its low solubility, and so is usually characterized as relatively immobile [22]. In contrast, the oxidized species of uranium, uranyl (U(VI), UO_2_^2+^), typically forms more stable aqueous complexes and is much more mobile in natural waters [22], although there are many known minerals containing uranium in the VI oxidation state [23].

### 3.3. Electrical Conductivity

Electrical conductivity (EC) is a measure of the ability of water to conduct an electrical current [24]. The range of EC at 25 °C in the water is 443–1358 µS/cm (Figure 3). The World Health Organization (WHO) recommends that the value of EC in water should not exceed 400 µS/cm. This means that the EC of the study area is higher than the normal threshold for EC in water. The average of the study area EC (785.1 µS/cm) is above the WHO recommended limit of EC in water (400 µS/cm).

Electrical conductivity is influenced by the availability of ions, and it is proportional to the total amount of dissolved solid, positively charged ions like sodium (Na), magnesium (Mg), calcium (Ca), iron (Fe) (cations) and chlorine (Cl), nitrate (NO_3_), sulphate (SO_4_), and phosphate (PO_4_)_,_ (anions) in the water. Electrical conductivity is also affected by water temperature; the warmer the water, the higher the conductivity [21]. Anions like Cl, NO_3_, SO_4_ are synonymous with sewage discharge. High concentrations of these anions were recorded at sites after the two water treatment plants and both sites have high EC (Figure 1C).

### 3.4. Anions Concentration

The concentrations of the anion (SO_4_, NO_3_, NO_2_, F, PO_4_, Cl_2_) relate to the source of water contamination. Typical sources for this kind of contamination are industrial operations, agricultural practices and mining activities, municipal waste discharge. The average concentration of SO_4_ (302.2 mg/L), NO_3_ (4.2 mg/L), NO_2_ (0.3 mg/L), F (0.1 mg/L) are below the WHO guideline limit (500 mg/L, 10 mg/L, 3 mg/L, 1.5 mg/L, respectively), only PO_4_ (1.3 mg/L) has a higher average concentration than the WHO guideline limit (0.03 mg/L). Using the South-African National Standard, SANS 241, the average of the SO_4_, NO_3_, NO_2_ and F in this study are all below the national limit.

The presence of sulphate, nitrate, nitrite, chlorine in water increases the salinity of the water. An increase in water salinity decreases the ability of U to be adsorbed by the sediment [15]. High concentrations of SO_4_, NO_3_ and Cl were recorded at sites after the two water treatment plants were established downstream of the Rietspruit (Figure 1C). At each of these sites, a low concentration of U was recorded in the sediment, despite the high concentration of U in the water column and the high adsorption capacity of the sediment.

### 3.5. Uranium Concentration in the Rietspruit Water

Within the immediate environment of the gold mine, the uranium concentration in the water is very high. At the inflow of the PW dam, a concentration of 781.9 µg/L was recorded in the furrow (Figure 4). This is about 200× higher than the WHO regulatory limit of U in water. The furrow takes discharge from the mine sewage plant to the PW dam. Another contributing factor to the high concentration of U at the furrow is the erosion of mine waste that litters the environment. During rain events, mine wastes are eroded into the furrow. In one of the mine wastes sampled next to a channel that connects with the furrow, U concentration of 1125 mg/kg was recorded in the sediment. Before the main rain event, the channel was dry, so the water sample collected after rainfall in the channel was analyzed. The concentration of U in the water was 387.7 µg/L. It confirms that erosion of mine wastes located on the earth surface could contribute to the increase in the concentration of U in water.

The discharge from the underground shaft has a concentration of 52.7 µg/L. It should be noted that, despite the treatment of fissure water before discharge in order to remove U, U still persists in the water. The Cold Lime Softening process is used by the mine to remove U in the water before discharge [25].

About 1 km from the dam inflow (Figure 4), the concentration of U has reduced to 181.2 µg/L. This site is the outflow of the PW dam. It shows the ability of the dam to remove U from the water column. This indicates immobilization of U. Further downstream, the concentration of U reduced further to 101.4 µg/L after another 1 km from the previous site (PW dam outflow). Reduction of U in water concentration as distance increases from the source pollutant has been reported [26]. This site is a wetland (Figure 1C). The reeds cover of the wetland and the hydraulic control point created as a result of the road culvert reduced the flow of the water. This allows for the deposition of suspended particles, which U is adsorbed to. The deposition resulted in having a high concentration of U in the wetland sediment, signaling immobilization of U by wetland sediment.

The reduction trend in the concentration of U continues downstream for another 6 km, where another dam is located (Figure 4). We call the dam a damlit because it does not have an official name. The concentration of U in the damlit water is 49.2 µg/L. After the damlit outflow, the concentration of U increased to 104.9 µg/L. This is because of the turbulence created by the damlit outflow slab. After the slab, the flow velocity of the stream increased to 177.9 m/s from 24.7 m/s recorded before the slab. This turbulence was able to resuspend fine particles of silt and slay which contains U, resulting in a high concentration of U in the water column.

The environment of the damlit is noted for a series of religious activities and farming. Many of the worshippers use the water for drinking, cooking and bathing, whereas the farmers use the damlit water for crop irrigation and animal watering. Due to the high concentration of U recorded in the water, these activities could be toxic to human health.

The concentration of U was consistent for another 2 km before a reduction in concentration. The reduction was caused by the diluting effect of the Ennerdale Water Treatment Plant (Figure 1C). The average discharge from the water plant averages about 132,300 m^3^ per month [25]. After the Ennerdale Water Treatment plant, there is a change in water color, from clear water to dark color, with a sewage odor. As a result of the discharge, the concentration of chlorine and nitrate in the water increased more than twofold. These increased the water salinity and reduced the adsorption capacity of the stream sediment [15]. The concentration of U after the Ennerdale Water Treatment plant was 70.2 µg/L; this value was consistent for another 2 km before the confluence of the Evaton Reitspruit tributary.

Evaton Rietspruit has no mining or industrial activity upstream (Figure 1C). As a result, the site was selected as this study background concentration of U. The concentration of U at the Evaton Rietspruit is 0.3 µg/L. This is the lowest concentration of U recorded in this study. Downstream of the Evaton Rietspruit is the Sebokeng Water Treatment Plant (Figure 1C), with an average monthly discharge of 2,191,867 m^3^ [25]. About 1 km from the Sebokeng Water Treatment Plant is the post confluence of Rietspruit and Evaton Rietspruit. The concentration of U in the water of this site was 4.9 µg/L. This is a huge decrease in U concentration. Before the confluence, 71.1 µg/L of U were recorded in the Rietspruit. The only reason for the reduction in U concentration is the diluting effect of the Sebokeng Water Treatment Plant. At a concentration of 4.9 µg/L, the concentration of U falls below the WHO regulatory limit in water.

The Leeuspruit tributary is an interesting tributary of the Rietspruit (Figure 1C). This is because of the gold mine established upstream. Uranium concentration in the Leeuspruit also reduces as the distance from the mining site increases. The highest concentration of U recorded in the Leeuspruit water is 50.2 µg/L. This is lower than expected because the mine is operational. At about 6.1 km before the confluence of the Leeuspruit and the Rietspruit, the concentration of U in the water has reduced to 12.81 µg/L. After the confluence, the concentration of U in the Rietspruit was 6.0 µg/L. This is an increase from the 4.9 µg/L U concentration recorded before the confluence of Rietspruit and Leeuspruit.

Going further downstream, the concentration of U ranges from 3.7 µg/L to 6.9 µg/L before the confluence with the Vaal River (Figure 1C). After the Vaal River confluence, about 60 km from the Rietspruit headwater, the concentration of U in the water was 3.0 µg/L. At another tributary selected as a natural background concentration of U in the water of the environment, the Klein Rietspruit (Figure 1C), the concentration of U recorded was 0.7 µg/L. Low concentration of U at the sites selected as the natural background concentration of U in the study area water (including the Evaton Rietspruit, 0.3 µg/L) indicates that the major source of U in the Rietspruit is as a result of the gold mine discharge and mine wastes.

### 3.6. Uranium Concentration in the Rietspruit Sediment

Gold mine wastes contain a high concentration of U. For example, in the slime dam, the concentration of U recorded was 72.7 mg/kg. A higher concentration of U has been recorded in slime dam in the country [27,28]. In another mine dump located beside a water channel that connects with the furrow taking sewage discharge to the PW dam, a concentration of 1043 mg/kg was recorded in the mine waste. Despite the decommissioning of the mine, the waste from mining operations contains a high concentration of U in them. The erosion of these mine wastes into nearby water bodies pollutes the stream, and this is toxic to downstream users of such water.

The scale found on the concrete canal, which takes discharge from the underground shaft, has a U concentration of 1143 mg/kg. The scale is formed as a result of calcium carbonate precipitation and changes in the atmospheric pressure [5]. The concentration of calcium in the scale is 20,760 mg/kg. This is the highest concentration of Ca found in this study. During the precipitation of calcium carbonate, U was also removed from the water column. The scale on the concrete canal could serve as another source of U as it could easily dissolve if the water chemistry should change. This could be influenced by a drop in the water pH, and also a change in the discharge flow. According to Winde et al. [28], the volume of water discharged by the mines at night and on the weekend is more than water discharged during the day. This is done in order to save costs spent on electricity.

Unlike the trend of U concentration in the Rietspruit water, which decreases with distance from the mining site, the trend of U in the Rietspruit sediment is not consistent (Figure 5). Several factors play an active role in the ability of sediment to retain U in them (immobilization). Some of these factors are sediment texture, organic carbon, water salinity, iron hydroxide/oxide, water flow velocity.

At the inflow of the PW dam, furrow, the concentration of U recorded was 611.1 mg/kg. All these concentrations were higher than the natural background concentration of U in soil (≤4 mg/kg by WHO). The percentage of organic carbon is very high (6.7%). Organic carbon has been reported to be able to keep an environment in a reducing condition. A reducing condition favors U immobilization. This is because U (VI), which is highly oxidized, soluble and mobile, will be reduced to U(IV), which is insoluble and immobile [13]. The correlation of U concentration in the sediment and organic carbon is positively strong (+0.7).

At the wetland located about 1.5 km from the PW dam inflow (Figure 5), the concentration of U at this site is 1237 mg/kg. This is the second highest concentration of U recorded in this study. The organic carbon plays a role in this site. The percentage of organic carbon here is 6.7%, which is similar to the Peter Wright inflow organic carbon. Uranium is adsorbed to organic matter, which breaks down to form humus. Furthermore, the high concentration of iron recorded at the wetland indicate co-precipitation of U with iron hydroxide. The correlation of U with OC and Fe is positively strong (+0.7 and +0.8 respectively). This indicates the importance of these two factors in the immobilization of U in the sediment.

The reed cover of the wetland was able to reduce the flow velocity of the wetland water. This resulted in the gravitational settling of fine particles of silt and clay. Fine particles of silt and clay have a higher adsorption capacity than the coarse particle of sand [15]. This is further established by the strong composition of silt and clay in the sediment, which amounts to a total percentage of 49% of the soil texture. Uranium adsorbed to fine suspended particles of clay and silt will precipitate out of the water column when the flow velocity of water has reduced and there are enough barriers (reeds in the case of the wetland) to impede the flow of the water. All of these factors contributed to the reason why a high concentration of U was recorded in the wetland sediment.

Wetlands have been categorized as a sink for U and other heavy metals [29,30,31]. Some of the concentration of heavy metals recorded in the wetland sediment are the highest found in these study (Table 2). Water discharged from the mine is the major reason why the stream is perennial. This could change if the discharge from the mine was stopped, which could lead to drying up of the stream during winter, thereby exposing the wetland sediment to atmospheric conditions. With the exposure, the sediment will become oxidized, and heavy metals stored in the sediment will be released during a rain event. This could be a major problem for downstream users of the stream.

The highest concentration of U was recorded in the damlit sediment (Figure 5). The concentration of U recorded was 1252 mg/kg. Low concentration of organic carbon was found in the sediment, which indicated that organic carbon does not play any role in this case. Possible factors that played a role here are the adsorption capacity of the damlit sediment and co-precipitation of U with iron and manganese hydroxide.

The total composition of silt and clay in the sediment is 31.1%. Similar to the wetland, this is a valid factor responsible for the immobilization of U in the damlit. In addition, the concentration of iron is very high, 30,630 mg/kg. As a result, co-precipitation of U with hardly soluble compound of iron is possible. Due to the reduction in the flow velocity of water in the damlit, precipitation of U of low solubility could be possible.

After the damlit, the concentration of U decreased to 16.5 mg/kg in the sediment of the next sampled site about 200 m from the damlit outflow. The reduction of U concentration in the sediment could be a result of the dominant coarse particle of sand. This resulted in a decrease in the ability of U to be retained in the sediment despite the high concentration of U found in the water (104.9 µg/L).

At the site after the Ennerdale Water Treatment Plant, the percentage of clay and silt is high, about 34% of the soil texture, but low concentration of U was recorded in the sediment. This is because of the increase in the salinity of the water, which reduces the adsorption capacity of the stream sediment. This finding is similar to the result of the Leeuspruit tributary: a similar tributary of the Rietspruit with an operational gold mine established upstream. The percentage of silt and clay in the Leeuspruit dam is high, above 40% at the inflow, middle and outflow of the dam. However, low concentration of U was found in the sediment. This is because of the water salinity. The concentration of chlorine, sulphate and nitrate in the water is very high.

Further downstream of the Rietspruit, the concentration of U was below 4 mg/kg, the natural concentration of U in soil [2] except at two sites (Figure 5), the post confluence of Leeuspruit and Rietspruit (5.6 mg/kg) and the inflow of the Lochvaal (15.2 mg/kg). The inflow of the Lochvaal was able to retain U as a result of the high composition of silt and clay of the sediment (above 50% of the soil texture) and the slow flow velocity of the water.

In the sediment of the sites chosen for this study of natural background concentration of U in sediment, the concentrations of U are 0.7 mg/kg at the Evaton Rietspruit and 0.6 mg/kg at the Klein Rietspruit. It further indicates that the environment has low concentration of U but, as a result of gold mining operations in the upstream environment, U was able to infiltrate the water and sediment of the Rietspruit.

## 4. Conclusions

The concentration of U decreases with increasing distance from the gold mine located in the study area. This reduction can be attributed to the water-sediment system interplay (chemistry), location of wetlands and dams close to the source pollutant.

The major source of U in the Rietspruit is the discharge from the Ezulwini plant underground shaft and water treatment plant. Other sources include the leaching of slime dams and mine dumps into the water channel leading to the PW dam. Underground pollution is another source of pollution mechanism, but not considered in this study. Despite the decommissioning of the mine, U trapped in mine wastes still contribute to the concentration of U in the water.

Wetland and dams provide a medium of filtering U out of water. However, this is not a permanent filtration, as U trapped in sediments can be remobilized back into the water column, thereby contributing to the pollution of the downstream. If the mine should stop discharging water into the PW dam, the stream will become non-perennial which will increase oxidation. Oxidation favors the remobilization of U trapped in sediment [5,13,15] when in contact with rainwater.

The impact of mining is not limited to the immediate environment. It is confirmed in this study that while high concentration of U is found in the water about 24 km from the mining site (Figure 4), highest concentration of U is found in the sediment about 6 km away from the mining site (Figure 5). As a result, animal grazing and religious activity along the stream must be discouraged within the environment with high concentration of U. The use of billboards and signage within the perimeter should be encouraged and effected. The use of the water from the dam for irrigation purpose should also be discontinued, as well as fishing activity in the dam.

Possible remediation of the mining environment, which must include the removal of the mine wastes that litter the mining sites, must be looked into. Also, possible mechanisms that will ensure the wetland chemistry does not change in order to prevent the remobilization of U from the sediment should be considered. This includes keeping the environment in a reduced state by maintaining the concentration of organic carbon and preventing the burning of wetland reeds during winter. In dams, the water level must be maintained to prevent drying out of the dam, which will result in the exposure of oxidized sediment to acid water from the rain.

## Figures and Tables

**Figure 1 ijerph-18-08466-f001:**
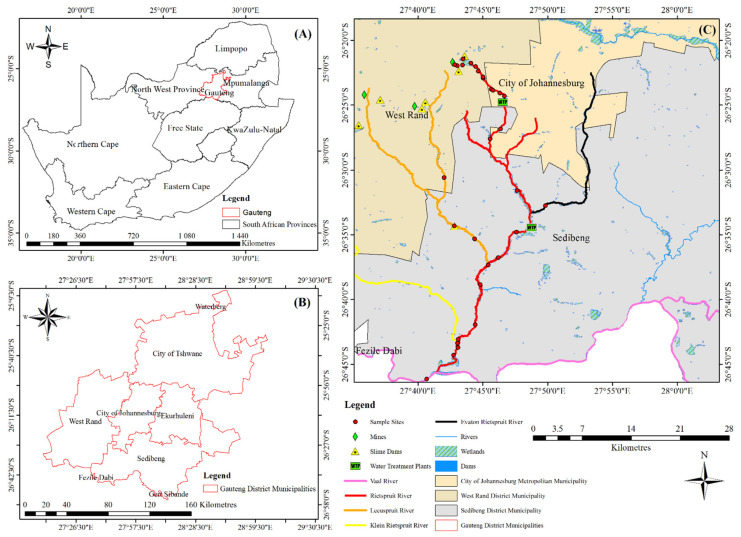
Map of study area. (**A**) shows the map of South Africa. (**B**) is the map of the Gauteng Province where the mine is located. (**C**) depicts the Rietspruit river with associated tributaries and the sampled sites.

**Figure 2 ijerph-18-08466-f002:**
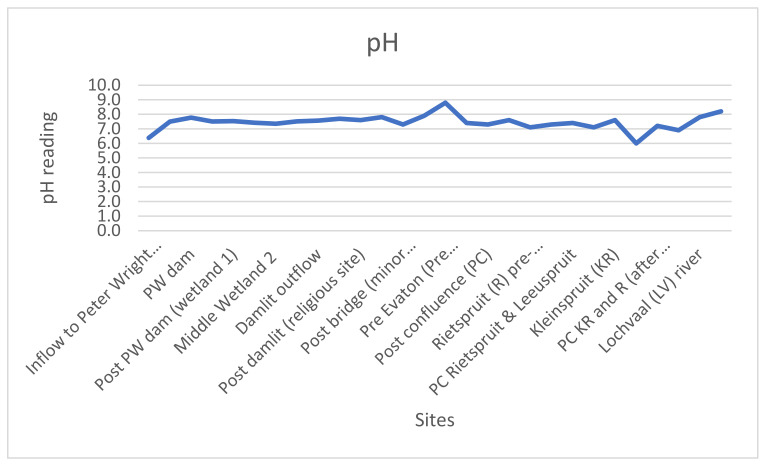
pH readings at the sampled sites.

**Figure 3 ijerph-18-08466-f003:**
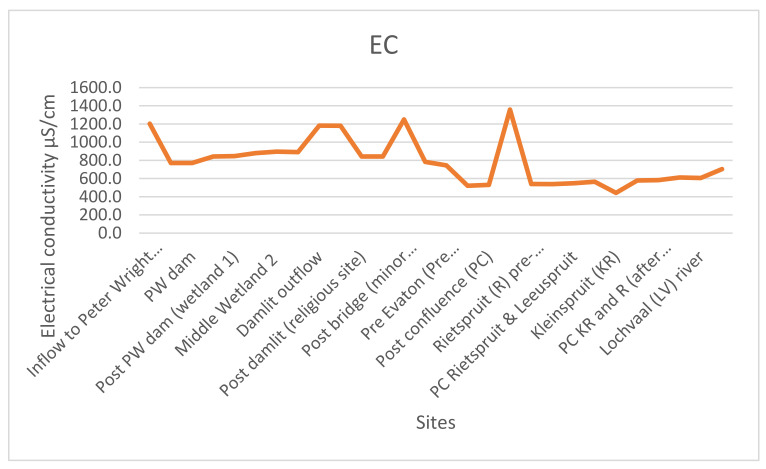
EC reading at the sampled sites.

**Figure 4 ijerph-18-08466-f004:**
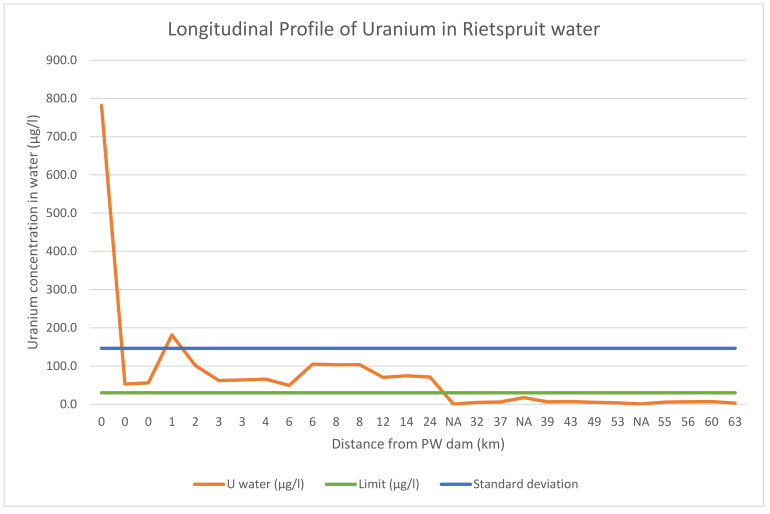
Longitudinal profile of U in Rietspruit water.

**Figure 5 ijerph-18-08466-f005:**
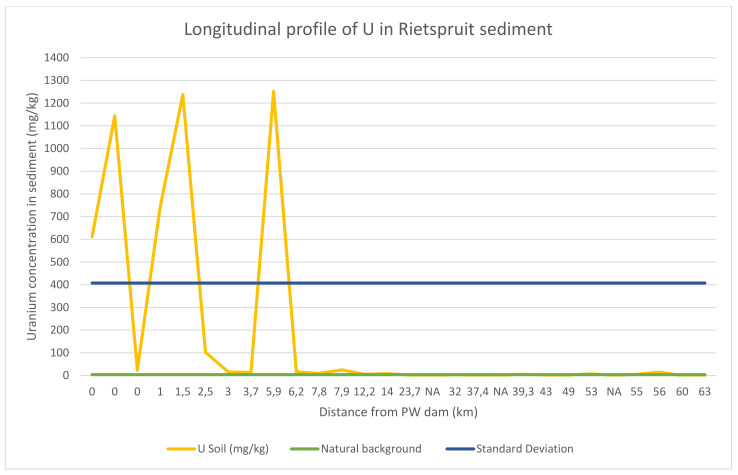
Longitudinal profile of U in Rietspruit sediment.

**Table 1 ijerph-18-08466-t001:** Rietspruit water chemistry.

Parameters	Water Samples (n = 28)					
	Maximum	Average	Minimum	SD	WHO Limit [19]	SANS 241 [20]
pH	8.8	7.4	6	0.7	6–9	5–9.7
EC (µS/cm)	1358	788	443	252	400	
Temp (°C)	26	16	8.8	4	30	
Cl (mg/L)	481.5	56	18.5	85	5	300
SO_4_ (mg/L)	1009	302.2	13.2	241	500	500
NO_3_ (mg/L)	25.1	4	0	6	10	11
NO_2_ (mg/L)	2.9	0	0	1	3	1
F (mg/L)	1.2	0.1	0	0.2	1.5	1.5
PO (mg/L)	14.8	1.1	0	2.8	0.03	

**Table 2 ijerph-18-08466-t002:** Concentration of heavy metals in wetland.

Heavy Metal	Gold	Arsenic	Lead	Cadmium	Mercury	Cobalt	Manganese
Concentration (mg/kg)	2.3	223.1	120.9	3.5	0.3	4878	47,120

## Data Availability

Not applicable.

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
