# Peer review of "Assessing Uranium Pollution Levels in the Rietspruit River, Far West Rand Goldfield, South Africa"

_ijerph, 2021, doi:10.3390/ijerph18168466_

Round 1
Reviewer 1 Report
This is a very important study that signifies the need for remediation of old mine tailings of gold deposits that can contain and release very high concentrations of uranium. Sometimes this concentration can easily reach 1000 ppm which could definitely be a problem for the environment. The authors identify a correlation between organic matter, iron and manganese which is also important to note. I am strongly for the publication of such research that brings up environmental problems. My comments are found in the pdf and can be easily addressed by the authors.

Author Response
Dear Reviewer,
Kindly find attached my response to all your comments.
Thanks.
Yours sincerely,
IB Raji

Reviewer 2 Report
The paper gives a good background on the problem of U in this area. There are a few typos and problems with word use and tenses. Often some words are not plural but should be and vice versa. There are a few typo (?) errors in the graphs for the numbers. I will attach a annotated pdf and include here line by line suggestions.
Reviewer A comments/observations and suggestions:
Line 38 user would be better as a plural users
Line 45 could use “for” instead of “in”
Line 47 delete “the”
Line 48 delete “which”
Line 49 delete “of” and ad an s to make “makes it crucial”
Line 51 delete “so as”
Line 60 missing reference.
Line 72 might list more than one mineral ( eg. pyrite, other minerals) and possibly a citation
Page 3 is blank?
Line 97 element might be better plural as “elements”
Line 108 -109 Table 1 some of the values should have decimal points not commas these are highlighted in yellow
Line 125 missing reference or wrong date? Winde (2003) there is a Winde (2010) in the references.
Line 129 & 130 could use a citation and better explanation. This statement is not exactly correct. Need to discuss the complexation state of the U. “The reduced species (U(lV), U4+) typically precipitates from natural waters near neutral pH due to its low solubility, and so is usually characterized as relatively immobile (Silva and Nitsche, 1995). In contrast, the oxidized species of uranium, uranyl (U(VI), UO22+), typically forms more stable aqueous complexes and is much more mobile in natural waters (Silva and Nitsche, 1995), although there are many known minerals containing uranium in the VI oxidation state (Finch and Murakami, 1999).”
Line 142 & 143 the cations and anions could have parenthesis around them instead of separated by commas sodium (Na), Calcium (Ca), etc.
Line 146 use “were” instead of “are”
Lines 168, 171, 184, 187 need to be consistent using PW or Peter Wright. Earlier in paper started using PW
Line 169 could use the number 200x instead of spelling it out
Line 171 Other should be Another
Line 172 waste is better than wastes and litter instead of “litters”
Line 187 increases rather than increase
Line 207 & 219 could use commas rather than spaces for the numbers
Line 210 delete “in”
Line 216 was selected for this study for the background
Line 226 The Leeuspruit tributary
Line 232 Is 6.0 a typo? Maybe this should be 60
Line 245 contain a high concentration of U.
Line 248 mine dump, located
Line 249 PW
Line 262 incomplete reference notation
Line 278 PW
Line 287 The reed cover
Line 289 clay have a higher adsorption
Line 294 All of these factors contributed
Line 296 Wetlands have been
Line 303 Is the rain acidic? Or is it that the rain will contribute to acid mine drainage because of the oxidized sediments and metals stored in the sediments?
Line 309 sediment and coprecipitation of U (delete comma and add “and”0
Line 318 delete as
Line 320 need decimal not comma (104.9 µg/l)
Line 336 use “for” instead of “as”
Line 342 decreases
Line 353 water column, thereby contributing
Line 369 This includes

Author Response
Dear Reviewer,
Thanks for taking the time to review my article.
I have gone through all the comments and implemented all the changes.
Thanks.
Yours Sincerely,
IB Raji
